# Genetic Variations of the *DPYD* Gene and Its Relationship with Ancestry Proportions in Different Ecuadorian Trihybrid Populations

**DOI:** 10.3390/jpm12060950

**Published:** 2022-06-10

**Authors:** Camila Farinango, Jennifer Gallardo-Cóndor, Byron Freire-Paspuel, Rodrigo Flores-Espinoza, Gabriela Jaramillo-Koupermann, Andrés López-Cortés, Germán Burgos, Eduardo Tejera, Alejandro Cabrera-Andrade

**Affiliations:** 1Facultad de Ingeniería y Ciencias Aplicadas, Universidad de Las Américas, Quito 170125, Ecuador; camilatw@hotmail.com (C.F.); jenniferg_nc@hotmail.com (J.G.-C.); eduardo.tejera@udla.edu.ec (E.T.); 2Laboratorios de Investigación, Universidad de Las Américas, Quito 170125, Ecuador; byron.freire@outlook.com (B.F.-P.); rodrigofabian.ffe@gmail.com (R.F.-E.); 3Vall d’Hebron Research Institute, Hospital Universitari Vall d’Hebron, 08035 Barcelona, Spain; 4Laboratório de Diagnóstico por DNA (LDD), Universidade do Estado do Rio de Janeiro, Rio de Janeiro 20550-013, Brazil; 5Laboratorio de Biología Molecular, Subproceso de Anatomía Patológica, Hospital de Especialidades Eugenio Espejo, Quito 170403, Ecuador; gaby_jaramillok@yahoo.com; 6Escuela de Medicina, Facultad de Ciencias de la Salud, Universidad de Las Américas, Quito 170125, Ecuador; aalc84@gmail.com (A.L.-C.); german.burgos@udla.edu.ec (G.B.); 7Programa de Investigación en Salud Global, Facultad de Ciencias de la Salud, Universidad Internacional SEK, Quito 170302, Ecuador; 8Latin American Network for the Implementation and Validation of Clinical Pharmacogenomics Guidelines (RELIVAF-CYTED), 28001 Madrid, Spain; 9Grupo de Bio-Quimioinformática, Universidad de Las Américas, Quito 170125, Ecuador; 10Carrera de Enfermería, Facultad de Ciencias de la Salud, Universidad de Las Américas, Quito 170125, Ecuador

**Keywords:** fluoropyrimidines, *DPYD*, single nucleotide variants, pharmacogenetics, ancestry analysis estimation, Ecuadorian ethnic groups

## Abstract

Dihydropyrimidine dehydrogenase is one of the main pharmacological metabolizers of fluoropyrimidines, a group of drugs widely used in clinical oncology. Around 20 to 30% of patients treated with fluoropyrimidines experience severe toxicity caused by a partial or total decrease in enzymatic activity. This decrease is due to molecular variants in the *DPYD* gene. Their prevalence and allelic frequencies vary considerably worldwide, so their description in heterogeneous groups such as the Ecuadorian population will allow for the description of pharmacogenetic variants and proper characterization of this population. Thus, we genotyped all the molecular variants with a predictive value for *DPYD* in a total of 410 Ecuadorian individuals belonging to Mestizo, Afro-Ecuadorian, and Indigenous ethnic groups. Moreover, we developed a genetic ancestry analysis using 46 autosomal ancestry informative markers. We determined 20 genetic variations in 5 amplified regions, including 3 novel single nucleotide variants. The allele frequencies for *DPYD* variants c.1627G>A (*5, rs1801159), c.1129-15T>C (rs56293913), c.1218G>A (rs61622928), rs1337752, rs141050810, rs2786783, rs2811178, and g.97450142G>A (chr1, GRCh38.p13) are significantly related to Native American and African ancestry proportions. In addition, the F_ST_ calculated from these variants demonstrates the closeness between Indigenous and Mestizo populations, and evidences genetic divergence between Afro-Ecuadorian groups when compared with Mestizo and Indigenous ethnic groups. In conclusion, the genetic variability in the *DPYD* gene is related to the genetic component of ancestral populations in different Ecuadorian ethnic groups. The absence and low frequency of variants with predictive value for fluoropyrimidine toxicity such as *DPYD* *2A, HapB3, and c.2846A>T (prevalent in populations with European ancestry) is consistent with the genetic background found.

## 1. Introduction

Dihydropyrimidine dehydrogenase (DPD) is a metabolic enzyme in the pyrimidine catabolic pathway, responsible for the conversion of thymine or uracil to 5,6-dihydrothymine or 5,6-dihydrouracil, respectively [1]. As a pharmacological metabolizer, it responds to the processing of drugs belonging to the fluoropyrimidine class, a group of drugs widely used in clinical oncology. The prodrug capecitabine and 5-Fluorouracil (5-FU) are two of the most widely used agents for the management of carcinoma of the colon, rectum, breast, stomach, and pancreas [2,3,4,5,6]. Once inside the system, only a small fraction (between 1 and 5%) is converted intracellularly into cytotoxic metabolites with chemotherapeutic characteristics, such as fluorodeoxyuridine monophosphate (FdUMP), fluorodeoxyuridine diphosphate (FdUDP), and fluorodeoxyuridine triphosphate (FdUTP). Of the remaining percentage, about 10% is excreted by the kidneys, and 80% of the administered drug is converted to an inactive metabolite (5,6-dihydro-5-fluorouracil, FUH2) by DPD [7]. Approximately 20–30% of cancer patients treated with fluoropyrimidines develop severe and potentially life-threatening toxicity early on during treatment [8]. According to the FDA, the toxicity caused by these drugs is distinguished by the development of mucositis, stomatitis or esophagopharyngitis, diarrhea (grade 3 or 4), palmar-plantar erythrodysesthesia (hand-foot syndrome), myelosuppression, hyperammonemic encephalopathy, and systemic toxicity. Toxicity grade 3 occurs in 25–30% of patients treated with this group of drugs [9,10], and is characterized by the development of neutropenia and leukopenia, vomiting, and skin ulceration [11,12]. Regarding systemic toxicity, in 1–18% of patients, one can observe the development of a spectrum of cardiac problems ranging from coronary ischemia, systolic left ventricular dysfunction, arrhythmias, and sudden death [13].

A decrease in the enzymatic activity of DPD is associated with the development of toxicity in patients treated with chemotherapeutic schemes based on fluoropyrimidines. Several studies have shown a partial deficiency of DPD in 3–8% of patients who develop toxicity. Of these, 50% showed a reduction in their enzymatic activity [14]. Molecular studies in patients with DPD deficiency have described more than 125 gene variants in the *DPYD* gene that result in partial or total loss of DPD activity [15], and the frequency of the molecular variants and the associated phenotypes appears to vary significantly between ethnic groups. For example, approximately 3–5% of the European population has a partial DPD enzyme deficiency and 0.1–0.2% has a complete DPD enzyme deficiency. On the other hand, about 8% of the African American population has a partial enzyme deficiency [16].

*DPYD* is one of the most studied genes in clinical oncology [12,17,18], where pharmacogenetic variants with high predictive values are described [19,20]. The *DPYD* single nucleotide variant (SNV) c.1905+1G>A (*2A, rs3918290) was the first to be described as functionally relevant [21]. Its allele frequency varies across different world populations, with very low and zero frequency in South Asian and Japanese (0.05% and 0%, respectively) populations, and slightly higher values in European-Caucasian (1.5%) and American-Caucasian (2.5%) populations [22,23,24,25]. The presence of c.1905+1G>A leads to a total deficiency in homozygous patients and a 50% reduction in enzyme activity in heterozygotes, increasing their risk of toxicity when treated with fluoropyrimidines [26,27]. Other *DPYD* variants such as c.2846A>T (rs67376798), c.1679T>G (*13, rs55886062), and c.1236G>A (single nucleotide variant belonging to *HapB3, rs56038477) were later identified as pharmacogenetic markers with clinical relevance [28]. In a systematic review that included 7356 patients from 8 different studies, a 4.4-fold relative risk of severe fluoropyrimidine-induced toxicity was found in carriers of the c.1679T>G (*13, rs55886062). In addition, a relative risk of 1.59 and 3.02 for severe toxicity was recorded for individuals carrying SNVs c.1236G>A and c.2846A>T, respectively [29].

Considering all this evidence, the Clinical Pharmacogenetics Implementation Consortium (CPIC) and the Dutch Pharmacogenetics Working Group (DPWG) [30,31] describe the nonfunctional and decreased variants as molecular markers with predictive value for cancer patients treated with fluoropyrimidines. Variants described as non-functional, such as *DPYD* c.1905+1G>A (*2A, rs3918290), c.1898del (*3, rs72549303), c.1679T>G (*13, rs55886062) and c.1156G>T (*12, rs78060119), and variants associated with a decreased function for the enzyme, such as c.2846A>T (rs67376798) and c.1236G>A (single nucleotide variant belonging to the *HapB3, rs56038477), are related to toxicity in carrier individuals that lies in a significant reduction in the enzymatic activity of DPD [25,29,32,33,34,35,36,37]. Thus, for carriers of the *DPYD* c c.1905+1G>A, c.1679T>G, c.2846A>T and c.1236G>A variants, an initial half dose of 50% of fluoropyrimidines is recommended [31]. It is also important to determine heterozygosis and homozygosis in the patient, since a total loss of DPD enzymatic activity is associated with severe toxicity. In these cases, alternative chemotherapy treatments are recommended. On the other hand, given the lack of association with toxicity in cancer patients, low penetrance, and/or contradictory results in clinical studies [33,38,39], the variants c.1896T>C (rs17376848), c.1627G>A (*5, rs1801159), c.1601G>A (*4, rs1801158), and c.85T>C (*9A, rs1801265) are described as fully functional.

Population genetic diversity is related to the inter-individual response to drugs used in pharmacological therapy [40]. One of the main problems of pharmacology and dosing charts lies in this variability, where the therapeutic response described for different types of treatments varies considerably across world populations [41,42]. The frequency of pharmacogenetic variants is heterogeneous throughout the world’s population [43,44], so its description in different ethnic groups shows the need for the inclusion of genetic screening methodologies. Reporting of predictive variants for *DPYD* in Ecuadorian heterogeneous populations will let us study both their genetic characterization, and above all, the implementation of personalized oncological therapies. The Ecuadorian population studied comprises three main ethnic groups: Mestizos, Afro-Ecuadorian, and Indigenous, distributed throughout the country. The genetic background for each group is different, hence it is paramount to consider their ancestry proportions and establish an association with the frequencies of the predictive variants. In fact, by determining the frequencies of these molecular variants in a healthy population, it will be possible to evaluate the usefulness of screening tests in the routine pretreatment of patients undergoing chemotherapy based on fluoropyrimidines, or will allow proposing alternative strategies for avoiding toxicity. The study population of this work represents the heterogeneity of the Ecuadorian people.

## 2. Materials and Methods

### 2.1. Sample Collection and DNA Extraction

A total of 410 unrelated Ecuadorian individuals (men and women) aged 18 and over were included in the study. Blood samples and buccal swabs were collected with written informed consent. The project was approved by the Ethical Committee in Human Research from the Universidad de las Américas (CEISH-UDLA 2017-0301).

The population studied consisted of Ecuadorian individuals from the 3 main geographic regions of the country: Coast (CO), Andes/Highlands (HG), and the Amazon Region (AZ). The ethnic self-denomination of the participants was also considered, namely Mestizo groups (MEZ), Afro-Ecuadorian (AFE), and Indigenous (IND). The MEZ groups inhabit the CO (Esmeraldas, Guayas, Los Ríos, and Manabí provinces), HG (Azuay, Bolivar, Carchi, Chimborazo, Cotopaxi, Imbabura, Loja, and Pichincha provinces), and AZ (Morona Santiago, Sucumbíos, and Zamora Chinchipe); the AFE inhabit the CO (Esmeraldas and Guayas) and HG (Carchi, Imbabura, and Pichincha); and IND groups inhabit the CO (Esmeraldas), HG (Bolivar, Carchi, Imbabura, and Pichincha provinces), and AZ (Napo, Orellana, and Pastaza provinces).

Genomic DNA extraction from blood samples was performed using the standard phenol–chloroform method [45]. For buccal swab samples, we used the resin Chelex 100 (Bio-Rad Laboratories, Hercules, CA, USA) at 10%. All samples were eluted in TE buffer to a final volume of 50 µL. The genetic material was quantified using a NanoDrop ND-1000 Spectrophotometer (NanoDrop Technologies, Willmington, DE, USA) at 260 nm. The quality was evaluated using the absorbance values at 280 and 230 nm. Finally, all samples were run on a 1% agarose gel to determine the degree of fragmentation.

### 2.2. Genotyping of DPYD SNVs

Genotyping was determined using Sanger sequencing, where the amplified PCR products were subsequently purified, analyzed, and aligned with the RefSeq NC_000001.11 (GRCh38.p13).

In order to design the primers, we used the Primer-BLAST tool (https://www.ncbi.nlm.nih.gov/tools/primer-blast/, accessed on 10 November 2022). We considered the default parameters and contemplated the inclusion of *DPYD* variants with clinical significance in oncological practice [30]. We selected 5 specific regions that include loci with the following characteristics: (1) variants described as nonfunctional for the DPD enzyme: *DPYD* c.1905+1G>A (*2A, rs3918290), c.1898del (*3, rs72549303), c.1679T>G (*13, rs55886062) and c.1156G>T (*12, rs78060119); (2) variants associated with a decreased function for the enzyme: c.2846A>T (rs67376798) and c.1236G>A (single nucleotide variant belonging to the *HapB3, rs56038477); and (3) variants described as fully functional for DPD: c.1896T>C (rs17376848), c.1627G>A (*5, rs1801159), c.1601G>A (*4, rs1801158), and c.85T>C (*9A, rs1801265).

The PCR reaction was performed using the GoTaq^®^ Green Master Mix (Promega, Madison, WI ,USA) according to the manufacturer’s instructions. Within the reaction, we used ~50 ng of DNA, 1x GoTaq Green Master Mix, and 0.3 µM of each primer, in a final volume of 25 µL. Following the amplification, the PCR products were purified using AMPure XP magnetic purification beads (Beckman Coulter, Brea, CA, USA), following the manufacturer’s directions, and were eluted in an elution buffer (Oxford Nanopore Technologies, Oxford, UK) for a final volume of 20 µL.

Next, the amplicons were sequenced using the BigDye^®^ Terminator v3.1 Cycle Sequencing Kit (Applied Biosystems, Austin, TX, USA) following the manufacturer’s instructions. Within the reaction, a final concentration of 1 µM of primer was used in a volume of 20 uL. The amplicons were purified using Agencourt CleanSEQ chemistry (Beckman Coulter, Brea, CA, USA), following the standard protocol, and ran in an ABI 3130 Genetic Analyzer 142 (Applied Biosystems, Foster City, CA, USA). The sequences were visualized using the GeneMapper software version 3.2 (Applied Biosystems, Foster City, CA, USA) and aligned in the Geneous program with the reference sequence NC_000001.11.

### 2.3. Genetic Ancestry Analysis

DNA samples were genotyped for a set of 46 autosomal ancestry informative markers (AIM-INDELs) in a single PCR multiplex reaction, applying the same previously described PCR conditions and primer concentrations [46]. Fluorescent DNA fragments were separated and detected by capillary electrophoresis using a 3130 Genetic Analyzer (Applied Biosystems) with Liz 600 and POP7 polymer. Data were analyzed and genotypes were automatically assigned with GeneMapper v5 (Applied Biosystems). The allele nomenclature handled is the same as previously described [46].

### 2.4. Statistical Analysis

Ancestry inferences were obtained using the admixture model with K = 3 (based in trihybrid historic admixture) on STRUCTURE V2.3.4 software [47]. Runs consisted of 100,000 burn-in steps, followed by 100,000 Markov chain Monte Carlo (MCMC) without taking into account group information a priori. All runs were made without any prior information on the origin of the samples. We only took into consideration the genetic background for the ancestral continental populations based on the European (EUR), African (AFR), and Native American (NAM) reference samples. This reference panel was extracted from the diversity panel of the Human Genome Diversity Project–Centre d’Etude du Polymorphisme Humain (HGDP–CEPH).

The estimated proportions of ancestral populations for each participant were compared with the genotyping results and geographic background using non-parametric tests (Kruskal–Wallis). The *p*-values < 0.01 were considered highly significant. Numeric variables were described using arithmetic means and standard deviations (SD). These statistical analyses were performed using the IBM SPSS Statistics 22 software (SPSS Inc., Chicago, IL, USA).

Finally, the population genetic parameters, such as allele frequencies, Hardy–Weinberg equilibrium (HWE), pairwise genetic distances (F_ST_), and analysis of molecular variance (AMOVA), were estimated using Arlequin software v.3.5.2.2 [48]. Regarding HWE analysis, the significance level of 0.05 was adjusted by applying Bonferroni’s correction. Considering the 20 SNVs found in the *DPYD* genotyping, *p*-values > 0.0025 show no significant deviations from HWE. For the analysis of pairwise, based on the classification proposed by Ballaux et al. (2002) F_ST_ values of 0–0.05 were interpreted as having little genetic distance between analyzed groups, values of 0.05–0.15 as moderate differentiation, and 0.15–0.25 as great differentiation [49]. The pairwise F_ST_ matrix was represented in multidimensional scaling (MDS) using the multidimensional scaling (PROXSCAL) of the IBM SPSS Statistics 22 software (SPSS Inc., Chicago, IL, USA).

## 3. Results

### 3.1. DPYD Genotyping and Hardy-Weinberg Equilibrium

We designed five sets of primers to amplify the *DPYD* regions that include the pharmacogenetics variants described in the methodology (Appendix A). From the genotyping results, the decreased variants c.1236G>A and c.2846 A>T show low allele frequencies for the minor allele (0.001 each), and we did not detect the nonfunctional alleles c.1905+1G>A, c.1898del, c.1679T>G and c.1156G>T. The fully functional variants c.1627G>A, c.85T>C, c.1896T>C, and c.1601G>A are more prevalent in the studied population, with allele frequency for the minor allele of 0.309, 0.174, 0.045, and 0.002, respectively. In addition, the applied methodology allowed us to screen additional variants within the amplified loci. The SNVs found and their calculated frequencies for the minor allele are: rs1672328735 (0.001), c.1129-15T>C (rs56293913) (0.034), c.1218G>A (rs61622928) (0.013), rs1337752 (0.729), rs141050810 (0.043), rs6657204 (0.006), rs2786783 (0.278), rs2811178 (0.65), rs76551168 (0.001), c.2846A>T (rs67376798) (0.001), rs978037367 (0.002), and rs1649194721 (0.001) (Figure 1).

Interestingly, we detected undescribed SNVs for *DPYD*: g.97450142G>A (chr1, GRCh38.p13), g.97450088T>G, and g.97082286A>C. The frequency of these SNVs for the minor allele was 0.001, except for the variant g.97450142G>A (0.059). The SNVs g.97450142G>A and g.97450088T>G are placed in the exonic region (NM_000110.3:c.1822C>T and NM_000110.3:c.1876A>C, respectively). Although g.97450142G>A is a synonymous variant (NP_000101.2:p.Leu608=), g.97450088T>G induces a change of threonine to proline at position 626 (NP_000101.2:p.Thr626Pro). The SNV at position g.97082286A>C is placed in the intronic region (NM_000110.3:c.2907+43T>G). All the allelic and genotypic frequencies, as well as the *p*-values calculated for HWE are described in Appendix A.

The HWE analysis showed no statistically significant deviations for most of the variants found in the Ecuadorian population, except for the SNVs c.85T>C, rs1337752, rs2786783, and rs2811178 (*p* < 0.0025). Nevertheless, when developing this analysis by grouping the individuals by ethnic self-denominations, only SNVs rs1337752 and rs2811178 did not meet the equilibrium in the MEZ and AFE groups, respectively. No statistically significant deviations from HWE expectations were detected for the 20 loci in the population under study when considering the groups by ethnic self-denominations and geographic location (Appendix A).

### 3.2. Ancestry Proportions in Ecuadorian Populations

The ancestry analysis estimation through the 46 AIM-INDELs describes the trihybrid composition of the Ecuadorian population studied, where the ancestry proportion of the NAM population (0.629) predominates, followed by AFR (0.228) and EUR (0.124) contributions. Considering ethnic groups, the self-proclaimed IND people show a high percentage of NAM ancestry (0.906), while in MEZ, the NAM and EUR proportions predominate in their genetic background (0.636 and 0.265, respectively). As expected, AFE populations show a high proportion of AFR ancestry (0.768) (Figure 2A).

The inference of ancestry proportions also describes the genetic composition of the ethnic groups throughout the three geographic regions studied. As can be seen in Figure 2B, a high percentage of NAM ancestry (0.883) is evidenced within the sampled provinces of the Amazon Region. On the other hand, the coastal region shows Ecuadorian groups with a significant degree of admixture between ancestry of NAM (0.420) and AFR (0.405) populations, while the Andes region indicates a pattern of admixture between NAM and EUR ancestry (0.565 and 0.189, respectively). The northern region of the country includes individuals with high percentages of AFR ancestry. Specifically, the provinces of Imbabura and Carchi (HG) and Esmeraldas province (CO) are those with the highest frequency of AFR ancestry throughout Ecuador (0.659, 0.309, and 0.590, respectively).

### 3.3. Relationship between DPYD Single Nucleotide Variants and Admixture Proportions in Ecuadorian Ethnic Groups

We compared the estimated proportions of each participant with the genotypes found for all SNVs with allele frequencies > 0.001 for the minor allele. We found that the variants c.1627G>A, c.1129-15T>C, c.1218G>A, rs1337752, rs141050810, rs2786783, rs2811178, and the SNV g.97450142G>A are significantly correlated with AFR ancestry. Similarly, these loci (except for c.1129-15T>C and c.1218G>A) show statistically significant differences with proportions calculated for NAM populations. None of the cases presented statistical differences when comparing the genotypes found with EUR ancestry (Appendix A).

### 3.4. DPYD Variants and Genetic Distances in the Ecuadorian Population

The data obtained from the genetic ancestry analysis and *DPYD* genotyping in the Ecuadorian population were used to estimate the F_ST_ genetic distances between all population pairs.

Firstly, the estimated F_ST_ genetic distances between all pairs of ancestral populations and Ecuadorian ethnic groups from the 46 AIM-INDELs showed statistically significant differences (*p* < 0.00000). Despite this, the low F_ST_ values indicate the closeness of IND and MEZ populations to NAM ancestral populations (F_ST_ = 0.019 and 0.065, respectively) and AFE groups to AFR ancestral populations (F_ST_ = 0.019). These values increase when comparing the ethnic groups with the EUR reference, having an F_ST_ of 0.275 with IND, 0.242 with AFE, and 0.145 with MEZ (Appendix A) populations. When comparing the F_ST_ values between the three Ecuadorian ethnic groups, the MEZ and IND populations show low divergence between them (F_ST_ = 0.0348), while the AFE population shows moderate differentiation (F_ST_ of 0.182 and 0.269, respectively). No intra-ethnic variation is shown when taking into account the geographical origin of the groups. The F_ST_ obtained in the IND of the three regions is low in all pairwise comparisons (F_ST_ < 0.0445). Likewise, the values obtained between the AFE groups of CO and HG, and the MEZ of the three regions are minimal (F_ST_ < 0.013 in all comparisons) (Figure 3A). This suggests that there is a genetic divergence between AFE populations with respect to MEZ and especially IND, but no substructure is evident when considering ethnic groups by region using the 46 AIM-INDELs.

AFR, EUR, and NAM continental ancestry groups are separated as three poles of diversity on the MDS plot of the pairwise F_ST_ values (Figure 3B). MEZ groups and especially IND are close to the NAM reference population, while AFE groups are close to AFR groups. The distance in the MDS plot from the EUR reference population shows its diminished contribution to the genetic background in the Ecuadorian population, where the greatest ancestral component is the NAM. The pairwise genetic distances (F_ST_) and the corresponding non-differentiation *p*-values are shown in Appendix A.

Secondly, from the genotyping results of the *DPYD* gene, we estimated the F_ST_ genetic distances between all population pairs. Similar to what was found with the 46 AIM-INDELs, the F_ST_ values showed little genetic distance between MEZ and NAM populations (F_ST_ = 0.00483, *p* = 0.0991); while comparing AFE groups with MEZ and IND, these values indicated moderate differentiation (F_ST_ = 0.06845 and 0.1069, respectively, *p* < 0.0000). When evaluating the ethnic groups by geographic origin, there is no great divergence, especially between MEZ groups of the three regions studied (F_STs_ = 0, *p* = 0), and in AFE groups of CO and HG (F_ST_ = 0.0038, *p* = 0.16216). The comparison of IND groups from the HG region with CO and AZ had low divergence (F_ST_ = 0.0371 and 0.0111, respectively, *p* = 0.14414); however, this index increased between IND groups from CO and AZ (F_ST_ = 0.0798, *p* = 0) (Figure 3C). All the pairwise genetic distances (F_ST_) and the corresponding non-differentiation *p*-values are shown in Appendix A. In the MDS plots of the pairwise F_ST_ values, the distribution of the points revealed that AFE populations (from HG and CO) have distinctive genetic profiles to both MEZ and IND geographic groups. Interestingly, IND groups from CO showed moderate differentiation with IND groups from AZ (Figure 3D).

Lastly, the differentiation test between all pairs of samples (Markov chain length: 100,000 steps) showed significant differences when comparing AFE with MEZ and IND groups (in both cases *p* < 0.001), while MEZ and IND groups did not show a statistically significant difference (*p* = 0.35860).

## 4. Discussion

The genotypic analysis applied to participants describes an Ecuadorian population that does not carry the nonfunctional variants *DPYD* c.1905+1G>A (*2A, rs3918290), c.1898del (*3, rs72549303), c.1679T>G (*13, rs55886062), and c.1156G>T (*12, rs78060119). The frequencies for the variants associated with a decreased function for the enzyme c.1236G>A (*HapB3, rs56038477) and c.2846A>T (rs67376798) are low (0.001). Conversely, the fully functional variants c.1896T>C (rs17376848), c.1627G>A (*5, rs1801159), and c.85T>C (*9A, rs1801265) are the most prevalent, with a frequency for the minor allele of 0.045, 0.301, and 0.194, respectively. Only the fully functional c.1601G>A (*4, rs1801158) registers a low frequency (0.002) in the tested population.

The prevalence of variants with phenotypic consequence in DPD or variants that are clinically associated with 5-FU toxicity are of very low frequency worldwide [50,51,52,53], including our country. The low prevalence found in the pharmacogenetic markers nonfunctional and decrease for *DPYD* suggests that genetic screening in Ecuadorian cancer patients treated with 5-Fu would not be the best strategy for avoiding toxicity. Alternative tests, such as measuring endogenous plasma dihydrouracil/uracil (UH2/U) ratios or measuring the DPD enzyme activity in peripheral blood mononuclear cells (PBMCs) [54,55,56], could be more informative as a way of avoiding complications in individuals treated with these chemotherapy regimens.

Despite the fact that fully functional variants were dismissed as having too low a penetrance to be of pharmacological importance, one should consider the reports in which a degree of association was found with secondary events during chemotherapeutic schemes based on fluoropyrimidines, especially with c.85T>C. The effect of c.85T>C on fluoropyrimidine treatment-related toxicity was initially reported in DPD deficient patients [57]; nonetheless, there are discrepancies about its phenotypic effect [58]. Individuals carrying the c.85C allele experienced diarrhea and hand-foot syndrome [59]. Moreover, researchers have reported a relationship between the c.85 T>C genotype and the development of grade 3–4 toxicities (diarrhea) in patients who received full doses of fluoropyrimidines [60]. Recently, a protective effect in carriers of the c.85C allele was described, suggesting greater DPD activity in individuals with this genotype [61,62]. Despite this association, one of the suggested limitations for the lack of association in clinical studies may be due to the linkage disequilibrium observed with c.496A>G (rs2297595) and c.1129-5923C>G (rs75017182) (two candidate markers known as risk allele for 5- FU toxicity) [63], so further evaluation of these variants in the context of clinical outcomes is suggested. In light of the fact that 19.4% of the Ecuadorian population is a carrier of the polymorphic allele, it is important to carry out future studies to evaluate the side effects in carriers of this variant, especially considering the recent reports on toxicity in oncology patients [64,65].

The Ecuadorian population shows a complex demographic history, for it is defined as multiethnic and multicultural. From the Spanish colonization onwards, the continuous admixture between Europeans, Native Amerindians [66], and people of African descent (the latter being mainly descendants of enslaved African people forced to enter the Americas through Caribbean ports) has shaped the patterns of diversity in the modern Ecuadorian population [67,68]. According to official data [69], 77.4% of the population call themselves Mestizo/a, 7.4% Montubio/a (mestizos belonging mainly to coastal regions), 7% Indigenous (that comprises 14 Indigenous nationalities scattered throughout the country), 6.1% white, 4.3% Afro-descendant, 1.9% Mulatto, 1% Black, and 0.4% other races. Despite this heterogeneity, the admixture proportions calculated from the 46 AIM-INDELs describe a trihybrid Ecuadorian population in which an ancestral NAM component predominates (Figure 2A), similar to that previously reported [70,71,72].

The IND groups show a high proportion of NAM ancestry (0.906), denoting their low degree of admixture with EUR and AFR populations. Ecuadorian MEZ groups evidence their history of admixture with the Spanish population, where the percentages of EUR (0.265) and NAM (0.636) ancestral proportions are prevalent in the individual genotypes. Although the largest contribution in MEZ is NAM, the *p*-value for the F_ST_ index is statistically significantly different (*p* = 0.0000), suggesting that the NAM component for an admixed population is heterogeneous [66,73]. The genetic distances evaluated by the F_ST_ (0.0348) show little genetic distance with IND groups, indicating a low degree of divergence between both groups and evidencing a high Native American contribution to their genetic ancestry.

While the NAM ancestry is the main composition of the MEZ, IND, and the Ecuadorian population considered as a whole, the AFR component is the predominant background for AFE groups (0.768). Afro-descendent populations in Ecuador have a discrete distribution, allocated along the Ecuadorian coastline, and mainly in the provinces of Esmeraldas and Imbabura. The estimated membership coefficients for AFR are in line with this, showing a higher frequency for this ancestral population in a relatively small territory in the north of the country (Figure 2B). This genetic composition in the population studied explains the low and/or null prevalence of deleterious variants in *DPYD*. The global frequencies for each of those nonfunctional and decreased variants, based on data from the Allele Frequency Aggregator (ALFA) project [74], ranges from 0.001 to 0.018. These frequencies differ among worldwide populations, where groups with European origin show a high prevalence (up to 0.02) when compared to populations of African (0.003) and South Asian origin (<0.001). Given the minor contribution of EUR ancestry in the Ecuadorian population, the absence of nonfunctional and decreased variants is consistent with the reported genetic background.

The genetic variability reported in *DPYD* is heterogeneous and has a close relationship with biogeographical ancestry [75,76,77,78]. The estimation of ancestry developed in the Ecuadorian ethnic groups allowed us to establish a relationship between the frequency of the minor alleles found in the different molecular variants of *DPYD* and the proportion of ancestry with ancestral reference populations, especially with NAM and AFR populations. *DPYD* variants c.1627G>A (*5, rs1801159), rs1337752, rs2786783, and rs2811178 have a statistically significant association with NAM ratio in genotyped individuals. Their frequency in individuals with MEZ and IND is higher than in AFE individuals. On the other hand, the c.1129-15T>C (rs56293913), c.1218G>A (rs61622928), rs141050810, and g.97450142G>A SNVs have an association with AFR ancestry, which shows differences in the genetic landscape and variant frequencies between ethnogeographic groups.

The mapping of variant frequencies in different ethnogeographic groups can provide important information about genetic structure, enabling better characterization of heterogeneous populations. In the HWE analysis, the loci c.85T>C (*9A, rs1801265), rs1337752, rs2786783, and rs2811178 showed statistically significant deviations; nevertheless, when stratifying the sample by considering ethnic self-identification and geographical origin, no statistically significant deviations from HWE expectations were detected. This imbalance can occur possibly due to geographic barriers that limit gene flow (migration) or genetic drift in subpopulations. Another possibility that cannot be ruled out is non-random mating or the existence of selection processes [79]. The excess of homozygotes calculated for the SNVs rs1337752, rs2786783, and rs2811178 (Appendix A) suggests a population substructure in the studied population. The genetic distances calculated from the variants found for *DPYD* describe genetic divergences between AFE populations when compared to NAM and MEZ. It is interesting to note that, as observed in the ancestry analysis estimation with the 46 AIM-INDELs, *DYPD* variants demonstrate the genetic closeness between MEZ and IND groups and show a divergence when compared to AFE populations.

It is important to note that the main objective of this study is to describe the prevalence and frequency of predictive variants in *DPYD* within a healthy Latin American population where their frequencies were not previously reported. One of the main limitations of this study is the lack of association between molecular variants in *DPYD* and toxicity in the cancer population treated with fluoropyrimidines. Since clinical and genetic factors can develop toxicity events to fluoropyrimidine-based chemotherapy, it is recommended to develop a prospective study on Ecuadorian patients treated with fluoropyrimidines and apply a more robust genetic screening using next generation sequencing. So far, the functional impact of several additional *DPYD* SNVs remains unknown, so a study in different ethnic groups may yield important results on molecular variants associated with decreased or lost function in DPD. On the other hand, even though they were included individuals belonging to the three main ethnic groups of Ecuador, distributed throughout the CO, HG and AZ regions, the number of participants studied does not allow for the evaluation of the intra-ethnic variation that exists in the Ecuadorian population. It is therefore necessary to increase the study population to evaluate minority groups within the ethnic groups analyzed.

## 5. Conclusions

In conclusion, the genetic diversity of the *DPYD* gene is heterogeneous within a multicultural and multiethnic Ecuadorian population. The frequency of the SNVs c.1627G>A (*5, rs1801159), c.1129-15T>C (rs56293913), c.1218G>A (rs61622928), rs1337752, rs141050810, rs2786783, rs2811178, and g.97450142G>A (chr1, GRCh38.p13) has a significant association with the genetic profile of Ecuadorian ethnic groups, especially with the component of ancestral NAM and AFR populations. The pairwise genetic distances calculated, both with variants in *DPYD* and with the 46 AIM-INDELs, demonstrate the closeness between IND and MEZ populations, and evidence genetic divergence between AFE groups when compared with MEZ and IND ethnic groups. In a trihybrid population where the NAM ancestry is the main genetic component, the absence and low frequency of predictive variants prevalent in European populations (c.1905+1G>A, c.1236G>A, and c.2846A>T) is consistent with the genetic profile found. Therefore, a genetic screening of variants with predictive value in *DPYD* would not be informative in Ecuadorian cancer patients treated with fluoropyrimidines. The use of alternative tests, for instance measuring endogenous plasma dihydrouracil/uracil (UH2/U) ratios or measuring the DPD enzyme activity in peripheral blood mononuclear cells (PBMCs), may be better strategies in order to avoid such toxicity.

## Figures and Tables

**Figure 1 jpm-12-00950-f001:**
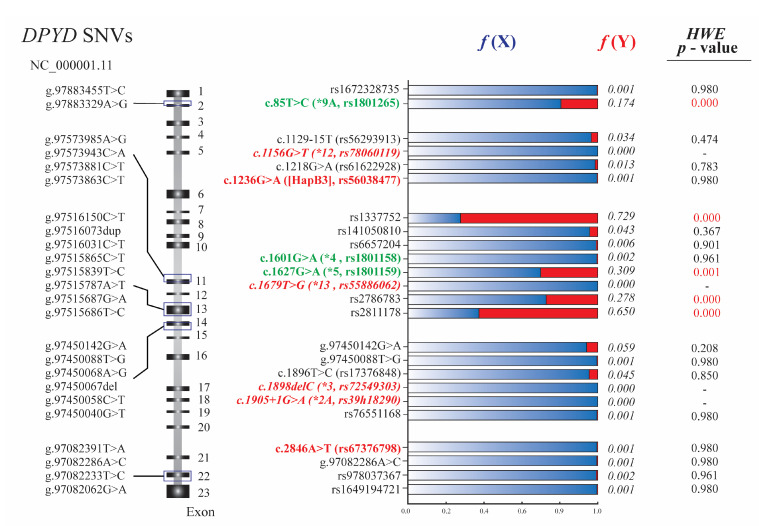
Schematic representation of the five amplified regions and the genetic variants found in the *DPYD* gene. The schematic represents the exonic regions (black boxes) and intronic regions of the DPYD gene. The positions of all the genetic variants found in the genotyping are described. Variants described as *nonfunctional* (red and italics), *decreased* (red and bold), and *fully functional* (green) are highlighted. The stacked bar chart represents the allele frequencies for each loci, where the blue color shows the frequencies for the wild-type allele *f*(X) and the red bars for the minor allele *f*(Y). The *p*-values found in the HEW analysis are also detailed. All values showing significant differences are marked in red.

**Figure 2 jpm-12-00950-f002:**
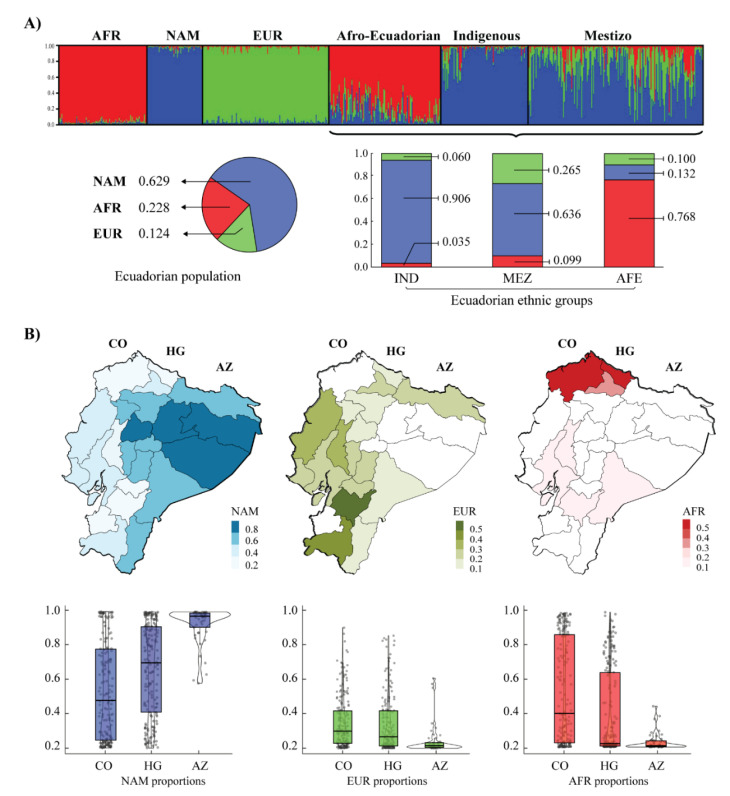
Analysis of ancestry proportions of the Ecuadorian population through 46 AIM-INDELs. (**A**) STRUCTURE Bar plot of the composition of the Ecuadorian ethnic groups compared with the reference population (AFR: Africa, EUR: Europe, and NAM: Native America) (k = 3, assuming migration model). While the pie chart shows the ancestry proportions of the total population studied, the stacked bar chart shows the composition of the different Ecuadorian ethnic groups (AFE: Afro-Ecuadorian, MEZ: Mestizo, and IND: Indigenous). (**B**) Distribution of the ancestry proportion of reference populations throughout Ecuador. The percentages for NAM, EUR, and MEZ ancestry in the Ecuadorian individuals studied are detailed separately. The box plot shows the composition of these proportions for Ecuadorian groups from the Coast (CO), Andes (HG), and Amazon Region (AZ).

**Figure 3 jpm-12-00950-f003:**
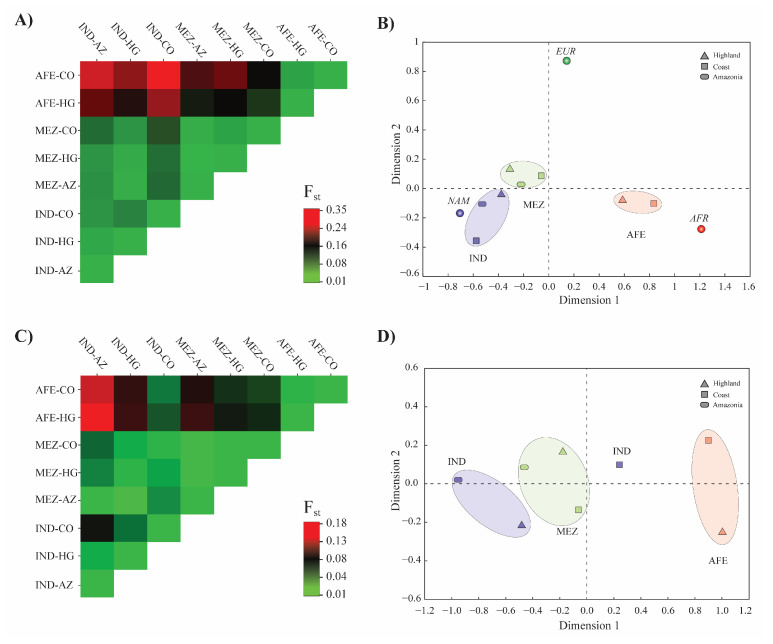
Genetic structure of the Ecuadorian ethnic groups evaluated by pairwise F_ST_ analyses. (**A**) Visualization of the matrix of pairwise F_ST_ values, and (**B**) multidimensional scaling analysis (MDS) based on F_ST_ genetic distances calculated from the 46 AIM-INDELs. (**C**) Visualization of the matrix of pairwise F_ST_ values, and (**D**) multidimensional scaling analysis (MDS) based on F_ST_ genetic distances calculated from the genetic variants found in *DPYD*. The groups analyzed correspond to Afro-Ecuadorian (AFE), Mestizo (MEZ), and Indigenous (IND) ethnic groups, belonging to the Coast (CO), Andes (HG), and Amazon Region (AZ).

## Data Availability

The data that supports the findings of this study are available in the Appendix A of this article.

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
