# Peer review of "Genetic Variations of the DPYD Gene and Its Relationship with Ancestry Proportions in Different Ecuadorian Trihybrid Populations"

_jpm, 2022, doi:10.3390/jpm12060950_

Round 1

Reviewer 1 Report

In the manuscript titled “Genetic variations of the DPYD gene and its relationship with ancestry proportions in different Ecuadorian tri-hybrid populations”, the authors genotyped all the molecular variants with predictive values for DPYD in a total of 410 Ecuadorian individuals belonging to Mestizo, Afro-Ecuadorian, and Indigenous ethnic groups. They found the genetic diversity of the DPYD gene is heterogeneous within a multi-cultural and multi-ethnic Ecuadorian population. This is a relatively new topic. The authors conduct research by abiding to best practices in method selection and high-quality research design and reporting results to the academic community. English writing is excellent.

Author Response

Many thanks for your comments.

Reviewer 2 Report

 I have several comments for the authors' consideration:

  1. Some more elaborate description on the motivation for this research would be helpful - i.e. why this specific population was chosen for this study and what is the degree of generalizability of the findings.
  2. A statement on the practical implications of this research may be sharpened - e.g., how can these findings help clinical researchers in oncology design better experiments/clinical trials? 
  3. The discussion should mention limitations of the current study - e.g. were there any confounding factors that could bias the results/conclusions?

A minor comment: line 189 - "mathematical means" is not correct; it should read "arithmetic means".

Reviewer 3 Report

Farinango  et al. investigate dihydropyrimidine dehydrogenase (encoded by the DPYD gene) in various Ecuadorian populations. This manuscript is of local interest.

Specific comments include:

  1. Please provide the rs number for  g.97450142G>A (chr1, GRCh38.p13).
  2. The English write-up is generally excellent but capitalization is at times idiosyncratic.
  3. The Discussion should be focused on the essential, ie it should be shortened substantially.
  4. Furthermore, the Discussion (if not elsewhere in the manuscript) must address the limited sample size of 410 individuals and implications thereof. A breakdown of the population may also be useful.

Round 2

Reviewer 3 Report

The authors have extensively revised some sections of their manuscript.

Farinango et al. have addressed many of the issues raised by this referee.

This manuscript is a resubmission of an earlier submission. The following is a list of the peer review reports and author responses from that submission.